# Computed tomography evaluation of skeletal muscle quality and quantity in people with morbid obesity with and without metabolic abnormality

Eunsun Oh[1☯], Nam-Jun Cho[2☯], Heemin Kang[1], Sang Hyun Kim[3], Hyeong Kyu Park[4], Soon Hyo Kwon[4]*

1 Department of Radiology, Soonchunhyang University Seoul Hospital, Seoul, Korea, 2 Department of Internal Medicine, Soonchunhyang University Cheonan Hospital, Cheonan, Korea, 3 Department of General Surgery, Soonchunhyang University Seoul Hospital, Seoul, Korea, 4 Department of Internal Medicine, Soonchunhyang University Seoul Hospital, Seoul, Korea

☯ These authors contributed equally to this work.
* ksoonhyo@gmail.com

**Data Availability Statement:** Yes - all data are fully available without restriction; All relevant data are

## Abstract

We investigated the differences in quantity and quality of skeletal muscle between metabolically healthy obese (MHO) and metabolically unhealthy obese (MUO) individuals using abdominal CT. One hundred and seventy-two people with morbid obesity who underwent bariatric surgery and 64 healthy control individuals participated in this retrospective study. We divided the people with morbid obesity into an MHO and MUO group. In addition, nonobese metabolic healthy people were included analysis to provide reference levels. CT evaluation of muscle quantity (at the level of the third lumbar vertebra [L3]) was performed by calculating muscle anatomical cross-sectional area (CSA), which was normalized to patient height to produce skeletal muscle index (SMI). Muscle quality was assessed as skeletal muscle density (SMD), which was calculated from CT muscle attenuation. To characterize intramuscular composition, muscle attenuation was classified into three categories using Hounsfield unit (HU) thresholds: -190 HU to -30 HU for intermuscular adipose tissue (IMAT), -29 to +29 HU for low attenuation muscle (LAM), and +30 to +150 HU for normal attenuation muscle (NAM). People with morbid obesity comprised 24 (14%) MHO individuals and 148 (86%) MUO individuals. The mean age of the participants was 39.7 ± 12.5 years, and 154 (65%) participants were women. MUO individuals had a significantly greater total skeletal muscle CSA than MHO individuals in the model that adjusted for all variables. Total skeletal muscle SMI, SMD, NAM index, LAM index, and IMAT index did not differ between MHO and MUO individuals for all adjusted models. Total skeletal muscle at the L3 level was not different in muscle quantity, quality, or intramuscular composition between the MHO and MUO individuals, based on CT evaluation. MHO individuals who are considered "healthy" should be carefully monitored and can have a similar risk of metabolic complications as MUO individuals, at least based on an assessment of skeletal muscle.

within the paper and its Supporting Information files.

**Funding:** S. H. K. received the funding from Soonchunhyang University Research Fund and Natioanl Resarch Foundation of Korea (NRF) funded by Medical Research Center (RS-2023-00219563). E. O. received Korean Health Industry Development Institute (KHIDI), funded by the Ministry of Health & Welfare (HI22C2193). The funders had no role in study design, data collection and analysis, decision to publish, or preparation of the manuscript.

**Competing interests:** The authors have declared that no competing interests exist.

# Introduction

Obesity is rapidly increasing worldwide. This phenomenon is associated with socioeconomic burden and poor clinical outcomes such as premature death and cardiovascular and other comorbidities [1]. However, some members of the obese population may have relatively favorable outcomes [2, 3]. Several studies have supported the concept of metabolically healthy obesity [2–4]. Whether metabolically healthy obesity exists is unclear, although the existence of a subgroup of metabolically healthy obese (MHO) individuals has been proposed [5–8]. MHO individuals are characterized by a lower cardiovascular risk and all-cause mortality [2–4].

Greater skeletal muscle fat infiltration is associated with higher all-cause and cardiovascular mortality [9]. Intramuscular fat content is associated with metabolic risk factors in the general population [10]. Skeletal muscle is the most significant organ for systemic glucose homeostasis and controls considerable insulin-stimulated systemic glucose uptake and release under normal conditions [11, 12]. In response to the impaired expandability of adipose tissue, excess lipids accumulate in skeletal muscle and result in insulin resistance in skeletal muscle, which is a major feature of type 2 diabetes mellitus and obesity [12, 13]. Body fat percentage is not different between MHO individuals and metabolically unhealthy obese (MUO) individuals when the groups are matched for body mass index (BMI) and sex [14]. However, metabolically healthy obesity is associated with a lower liver fat levels and lower skeletal muscle fat infiltration, compared to metabolically unhealthy obesity [15]. Less steatosis in the organs of MHO individuals suggests a mechanism of greater insulin sensitivity in these individuals than in MUO individuals [16].

Computed tomography (CT) has been commonly used in the evaluation of muscle in numerous studies [17–20]. With CT, measuring muscle quantity and quality (e.g., myosteatosis) is possible [19, 21, 22]. CT evaluation of muscle quantity is performed by calculating muscle cross-sectional area (CSA), which is normalized for patient height, thereby producing skeletal muscle index (SMI) [21]. Muscle quality evaluated on CT is based on muscle attenuation and is expressed as skeletal muscle density (SMD) [21]. Increased fat accumulation in muscle (i.e., myosteatosis) is characterized by a lower attenuation of muscle on CT images [21]. In addition, skeletal muscle areas can be classified as normal attenuation muscle (NAM), low attenuation muscle (LAM), and intermuscular adipose tissue (IMAT), based on Hounsfield unit (HU) thresholds on CT [23]. Several studies have recently been conducted on the association between skeletal muscle and metabolic syndrome using CT [24, 25]. In addition, muscle quality and quantity of skeletal muscle was measured between MHO and MUO in general population [26]. However, no studies exist on the relationship between metabolic risk-related subtypes in the people with morbid obesity (i.e., MUO and MHO) and skeletal muscle, evaluated by CT.

Therefore, the purpose of our study was to determine the difference in the quantity and quality of skeletal muscle between MHO and MUO individuals by using CT to demonstrate the existence of specific subtypes among the people with morbid obesity such as MHO and MUO individuals. We hypothesized that quality or quantity of skeletal muscle should differ between MHO and MUO individuals.

# Methods

## Patients

We collected data from people with morbid obesity who underwent bariatric surgery between October 2009 and February 2019. Nonobese healthy control individuals were kidney donor candidates from November 2003 to February 2019. Individuals were excluded who had any

cancer, heart failure, liver cirrhosis, chronic kidney disease, active infection, or serious cardiovascular disease. The people with morbid obesity and nonobese healthy controls underwent abdominal and pelvic CT scans. We reviewed the electronic medical records of each participant. We accessed the data from January through December 2022. The participants were categorized based on BMI and the presence of metabolic syndrome. Obesity was defined as a BMI of $\geq$30 kg/m$^2$. Metabolic syndrome was defined based on the presence of three or more metabolic syndrome components using the Adult Treatment Panel III Asian criteria [27]: waist circumference $\geq$90 cm in men or $\geq$80 cm in women; a fasting triglyceride (TG) level $\geq$150 mg/dL or treatment for elevated TG; a high-density lipoprotein (HDL) cholesterol level <40 mg/dL in men or <50 mg/dL in women or drug treatment for low HDL cholesterol level; systolic blood pressure $\geq$130 mmHg, diastolic blood pressure $\geq$85 mmHg, use of antihypertensive medication, or history of hypertension; and fasting glucose $\geq$100 mg/dL or a previous diagnosis of type 2 diabetes. The participants were classified as nonobese metabolically healthy individuals, MHO individuals, or MUO individuals. This study is reported as per the Strengthening the Reporting of Observational Studies in Epidemiology (STROBE) guideline (S1 Table).

## CT examination protocol

All participants underwent abdominal and pelvic CT scans with one of three 64-slice scanners (SOMATOM Definition Edge, Siemens Medical Solutions, Erlangen, Germany; SOMATOM Sensation 64, Siemens Medical Solutions, Erlangen, Germany; and Discovery CT750 HD; GE Healthcare, Milwaukee, WI, USA). Tube voltages of 100 kV, 120 kV, or 140 kV and a tube current of 250–300 mA were used. The section collimation was 64 mm × 0.6 mm, and the slice thickness was 3.0 mm, 3.75 mm, or 5.0 mm in 3-mm or 5-mm increments. The gantry rotation time was 50 ms.

## CT image measurements

A musculoskeletal radiologist with 5 years of experience following fellowship training, who was blinded to the clinical data, independently made the measurements on each CT scan. Measurements were conducted using commercial 3D analysis software (Aquarius iNtuition, v4.4.12; TeraRecon, Foster City, CA, USA) (Fig 1. Representative images of characterization of intramuscular composition. Skeletal muscle attenuation was classified into three categories using Hounsfield unit (HU) thresholds: -190 to -30 HU for intermuscular adipose tissue (IMAT), -29 to +29 HU for low attenuation muscle (LAM), and +30 to +150 HU for normal attenuation muscle (NAM). (A) A 51-year-old man from the nonobese healthy control group. (B) A 56-year-old man from one of the obese groups.). To assess muscle quantity and quality, muscle cross-sectional area (in centimeters squared) and mean muscle attenuation (in HU) were measured on pre-contrast axial images. The radiologist drew freehand regions of interest around the circumference of the psoas, paraspinal, and abdominal wall skeletal muscle groups at the pedicle level of the third lumbar vertebra (L3). The software automatically calculated the cross-sectional area and mean skeletal muscle attenuation. The cross-sectional areas of the muscle groups were summed and divided by patient height (in meters squared) to calculate SMI [20, 28]. SMD was calculated as the mean attenuation of the muscle groups at the L3 level. To characterize intramuscular composition, muscle attenuation was classified into three categories, using HU thresholds: -190 to -30 HU for IMAT, -29 to +29 HU for LAM, and +30 to +150 HU for NAM [23, 29]. The cross-sectional areas of each HU range for all muscle groups were calculated and normalized for patient height and are expressed as IMAT, LAM, and NAM indices. In the research setting, a single cross-sectional abdominal image from a

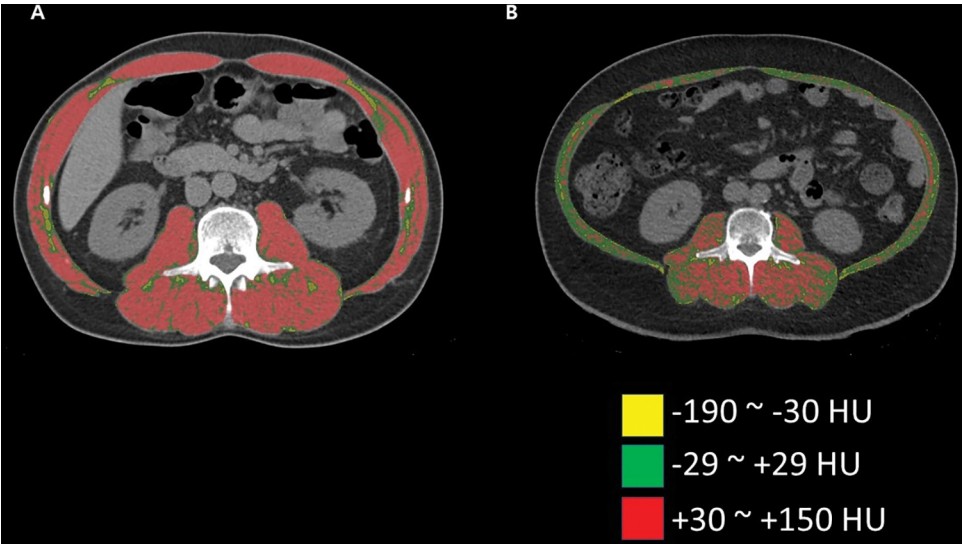

**Fig 1.**

diagnostic CT has been used to assess abdominal circumference, abdominal adipose tissue and skeletal muscle areas. A single cross-sectional CT image of the L3 region can accurately estimate body composition, including regional abdominal adipose tissue, skeletal muscle and waist circumference [30].

The reliability of the measurements was evaluated by reassessing 78 CT images twice: once by the same reader (after 2 years washout period, intrareader) and once by a second reader (a radiologist with resident training, inter-reader). On the basis of the results of a similar previous study [31] (intraclass correlation coefficient = 0.93), it was calculated that a sample size of 27 was required to detect a good intraclass correlation coefficient with a power of 80% and an $\alpha$ of 0.05. We randomly selected 27 CT images each from the control and MUO groups and all 24 CT images from the MHO group.

## Statistical analysis

Categorical variables are expressed as the count (percentage). Normally distributed continuous variables are expressed as the mean ± the standard deviation. Non-normally distributed continuous variables are presented as the median (interquartile range). The distribution of the variables was visually assessed through a histogram and quantile-quantile plot. The comparisons between MHO group and MUO group were analyzed using Student's t-test, Welch's t-test or Mann-Whitney U test, as appropriate. We used Levene's test to assess the equality of variances between the two groups. The Spearman's rank correlation coefficient was used to compare the categorical variables. The Pearson's correlation coefficient was used to test the correlation between individual continuous variables. Multiple linear regression was conducted to identify any association between the metabolic groups and the fat index values, after adjusting for possible confounders. The IMAT index was logarithmically transformed because of its right-skewed distribution. The intrareader and inter-reader reliabilities of this measurement were assessed by reporting the intraclass correlation coefficient and 95% confidence interval. Statistical significance was set at $p < 0.05$. Two-tailed tests were conducted for all hypothesis tests. Statistical analyses were conducted using R (version 3.6.1; R Foundation for Statistical Computing, Vienna, Austria).

## Results

### Participant characteristics

The study enrolled 172 obese participants, comprising 24 MHO participants and 148 MUO participants, and 64 nonobese metabolically healthy participants (82 men and 154 women). The mean age of the participants was 39.7 ± 12.5 years. The clinical characteristics of participants are presented in Table 1. MUO individuals were significantly older and heavier than MHO individuals, but BMI and waist circumference were not different between the two groups. The prevalence of diabetes mellitus and hypertension was higher in MUO individuals than in MHO individuals. Blood pressure, serum glucose, glycated hemoglobin, aspartate aminotransferase, alanine aminotransferase, and TG levels were also higher in MUO individuals than in MHO individuals, whereas HDL cholesterol was lower in MUO individuals (Table 1).

**Table 1. Baseline characteristics of the study participants.**

| Parameter | Control (n = 64) | MHO (n = 24) | MUO (n = 148) | *p value* MHO vs MUO |
|---|---|---|---|---|
| Age (years) | 42.8 ± 13.3 | 32.9 ± 12.2 | 39.4 ± 11.8 | 0.013* |
| No. of women [i] | 35 (54.7) | 21 (87.5) | 98 (66.2) | 0.054 |
| Systolic blood pressure (mmHg) | 114.4 ± 10.9 | 122.0 ± 9.2 | 130.3 ± 15.5 | < 0.001* |
| Diastolic blood pressure (mmHg) | 72.2 ± 8.5 | 71.0 ± 8.4 | 76.0 ± 10.5 | 0.027* |
| Height (cm) | 164.1 ± 9.4 | 162.5 ± 6.7 | 166.4 ± 9.7 | 0.019* |
| Weight (kg) | 63.2 ± 10.0 | 101.8 ± 17.1 | 111.1 ± 23.2 | 0.022* |
| Body mass index (kg/m$^2$) | 23.4 ± 2.8 | 38.4 ± 5.0 | 39.8 ± 6.3 | 0.252 |
| Waist circumference (cm) | 79.2 ± 8.7 | 111.0 ± 12.8 | 115.6 ± 12.7 | 0.086 |
| Hemoglobin (g/dL) | 13.2 ± 1.6 | 13.4 ± 1.0 | 13.8 ± 1.6 | 0.144 |
| Platelet (10$^3$/mm$^3$) | 229.9 ± 57.3 | 289.3 ± 67.4 | 264.1 ± 64.3 | 0.083 |
| Total protein (g/dL) | 6.7 ± 0.7 | 7.1 ± 0.5 | 7.1±0.5 | 0.908 |
| Albumin (g/dL) | 4.1 ± 0.5 | 4.3 ± 0.3 | 4.3 ± 0.4 | 0.677 |
| Serum glucose (mg/dL) [ii] | 107.5 (89.8,130.0) | 95.5 (89.8,102.0) | 116.5 (101.0,155.2) | <0.001* |
| Total bilirubin (mg/dL) [ii] | 0.6 (0.4, 0.7) | 0.5 (0.4, 0.62) | 0.5 (0.4, 0.7) | 0.811 |
| BUN (mg/dL) | 13.6 ± 3.7 | 11.6 ± 3.0 | 14.1 ± 4.8 | 0.013* |
| Cr (mg/dL) | 0.86 ± 0.19 | 0.68 ± 0.14 | 0.75 ± 0.23 | 0.141 |
| eGFR (mL/min) | 95.6 ±17.1 | 115.0 ± 13.0 | 106.6 ± 19.1 | 0.045* |
| AST (IU/L) [ii] | 20.0 (17.0, 23.0) | 28.0 (22.0, 52.0) | 46.5 (28.0, 73.2) | 0.025* |
| ALT (IU/L) [ii] | 15.0 (11.0, 21.0) | 31.0 (20.8, 58.2) | 54.0 (33.8, 95.5) | 0.007* |
| Total cholesterol (mg/dL) | 181.4 ± 36.4 | 198.6 ± 33.6 | 189.8 ± 47.8 | 0.384 |
| HDL (mg/dL) | 57.1 ± 12.9 | 58.3 ± 13.8 | 43.5 ± 10.7 | < 0.001* |
| LDL (mg/dL) | 110.2 ± 27.8 | 131.0 ± 30.2 | 122.0 ± 43.8 | 0.326 |
| TG (mg/dL) [ii] | 82.5 (63.8, 119.5) | 99.5 (83.0, 123.8) | 183.0 (139.8, 254.0) | < 0.001* |
| HbA1c (%) [ii] | 5.5 (5.10, 5.8) | 5.5 (5.35, 5.6) | 6.5 (5.7, 7.8) | < 0.001* |
| Current Smoking [i] | 9 (14.1) | 2 (8.3) | 43 (29.1) | 0.043* |
| Alcohol consumption [i] | 21 (32.8) | 8 (33.3) | 47 (31.8) | 1.000 |
| Lipid medication [i] | 2 (3.1) | 0 (0.0) | 44 (29.7) | <0.001* |

Note. Except where indicated, the data are mean ± standard deviation. MHO, metabolically healthy obese; MUO, metabolically unhealthy obese; BUN, blood urea nitrogen; Cr, creatinine; eGFR, estimated glomerular filtration rate; AST, aspartate aminotransferase; ALT, alanine aminotransferase; HDL, high-density lipoprotein; LDL, low-density lipoprotein; TG, triglyceride; HbA1c, hemoglobin A1c.

* *p value* was statistically significant.

i. Data are numbers. The data in parentheses are percentages.

ii. Data are medians. The data in parentheses are the first and third quartiles.

**Table 2. Correlations between clinical parameters and number of metabolic syndrome components in MHO and MUO participants.**

| Variable | Correlation with number of metabolic syndrome components | | | | | |
|---|---|---|---|---|---|---|
| | Total (n = 172) | | Men (n = 53) | | Women (n = 119) | |
| | r | p value | r | p value | r | p value |
| Age (years) | 0.174 | 0.022* | 0.144 | 0.303 | 0.200 | 0.029* |
| Systolic blood pressure (mmHg) | 0.276 | 0.000* | 0.220 | 0.114 | 0.290 | 0.001* |
| Diastolic blood pressure (mmHg) | 0.206 | 0.007* | 0.158 | 0.259 | 0.224 | 0.015* |
| Height (cm) | 0.038 | 0.620 | 0.165 | 0.238 | -0.017 | 0.858 |
| Weight (kg) | 0.036 | 0.644 | -0.124 | 0.376 | 0.005 | 0.953 |
| Body mass index (kg/ m$^2$) | 0.010 | 0.899 | -0.284 | 0.039* | 0.052 | 0.572 |
| Waist circumference (cm) | 0.083 | 0.278 | -0.098 | 0.483 | 0.077 | 0.403 |
| Hemoglobin (g/dL) | 0.027 | 0.729 | -0.044 | 0.756 | 0.016 | 0.862 |
| Platelet (10$^3$/mm$^3$) | -0.072 | 0.350 | -0.222 | 0.110 | -0.001 | 0.987 |
| Total protein (g/dL) | 0.008 | 0.921 | -0.216 | 0.121 | 0.094 | 0.309 |
| Albumin (g/dL) | 0.095 | 0.214 | -0.095 | 0.497 | 0.172 | 0.061 |
| Serum glucose (mg/dL) | 0.462 | 0.000* | 0.468 | 0.000* | 0.463 | 0.000* |
| Total bilirubin (mg/dL) | -0.084 | 0.272 | -0.131 | 0.350 | -0.099 | 0.285 |
| BUN (mg/dL) | 0.104 | 0.174 | 0.083 | 0.554 | 0.100 | 0.278 |
| Cr (mg/dL) | 0.010 | 0.893 | -0.044 | 0.756 | -0.020 | 0.831 |
| eGFR (mL/min) | -0.118 | 0.124 | -0.076 | 0.587 | -0.132 | 0.153 |
| AST (IU/L) | 0.129 | 0.092 | 0.004 | 0.979 | 0.175 | 0.056 |
| ALT (IU/L) | 0.130 | 0.090 | -0.070 | 0.620 | 0.204 | 0.026* |
| Total cholesterol (mg/dL) | -0.070 | 0.363 | -0.057 | 0.685 | -0.072 | 0.437 |
| HDL (mg/dL) | -0.563 | 0.000* | -0.514 | 0.000* | -0.599 | 0.000* |
| LDL (mg/dL) | -0.092 | 0.234 | -0.154 | 0.290 | -0.071 | 0.445 |
| TG (mg/dL) | 0.556 | 0.000* | 0.445 | 0.001* | 0.605 | 0.000* |
| HbA1c (%) | 0.466 | 0.000* | 0.453 | 0.001* | 0.474 | 0.000* |

Note. BUN, blood urea nitrogen; Cr, creatinine; eGFR, estimated glomerular filtration rate; AST, aspartate aminotransferase; ALT, alanine aminotransferase; HDL, high-density lipoprotein; LDL, low-density lipoprotein; TG, triglyceride; HbA1c, hemoglobin A1c.

* *p value* was statistically significant.

Correlations between clinical parameters and number of metabolic syndrome components in participants with MHO and MUO are presented in Table 2. In men and women, clinical parameters such as serum glucose ($r = 0.462$, $p = 0.000$), HbA1c ($r = 0.466$, $p = 0.000$), HDL ($r = -0.563$, $p = 0.000$), and TG ($r = 0.556$, $p = 0.000$) were correlated with metabolic syndrome components. In only women, systolic blood pressure ($r = 0.290$, $p = 0.001$) and diastolic blood pressure ($r = 0.224$, $p = 0.015$) were positively associated with number of metabolic syndrome components. However, no association was found between waist circumference and number of metabolic syndrome components.

## Reliability of the measurement

Measurement for muscle imaging parameters were shown to have excellent intrareader and inter-reader reliabilities (intraclass correlation coefficient, 0.91–0.99; S1 Data).

## Imaging parameters

Table 3 shows the imaging parameters for each group and differences in imaging parameters between MHO and MUO individuals after adjustment. MUO individuals had a significantly

**Table 3. Muscle imaging parameters in healthy controls and MHO and MUO participants.**

| Total muscle | Control (n = 64) | MHO (n = 24) | MUO (n = 148) | MHO vs MUO (p value) | | |
|---|---|---|---|---|---|---|
| | | | | Model 1 | Model 2 | Model 3 |
| CSA ($cm^2$) | 143.5 ± 20.7 | 196.7 ± 17.4 | 202.5 ± 25.0 | 0.019* | 0.042* | 0.041* |
| SMI [$cm^2/(m^2)$] | 48.7 ± 7.6 | 69.8 ± 3.9 | 65.3 ± 7.8 | 0.084 | 0.131 | 0.129 |
| SMD (HU) | 44.0 ± 7.2 | 27.6 ± 3.0 | 29.9 ± 9.5 | 0.620 | 0.628 | 0.466 |
| NAM index [$cm^2/(m^2)$] | 40.2 ± 6.7 | 30.3 ± 2.9 | 30.9 ± 8.7 | 0.475 | 0.473 | 0.653 |
| LAM index [$cm^2/(m^2)$] | 8.3 ± 5.3 | 25.2 ± 8.9 | 23.7 ± 9.1 | 0.086 | 0.088 | 0.070 |
| IMAT index [$cm^2/(m^2)$] [i] | 1.44 (0.60, 1.88) | 6.49 (5.24, 6.83) | 4.70 (2.69, 7.11) | 0.531 | 0.557 | 0.410 |

Note. Unless otherwise indicated, data are presented as the mean ± standard deviation. MHO, metabolically healthy obese; MUO, metabolically unhealthy obese; CSA, cross-sectional area; SMI, skeletal muscle index; SMD, skeletal muscle density; HU, Hounsfield unit; NAM, normal attenuation muscle; LAM, low-attenuation muscle; IMAT, intermuscular adipose tissue. Model 1 = multivariate analysis adjusted for age and sex, Model 2 = multivariate analysis adjusted for age, sex, current smoking status, and alcohol consumption, Model 3 = multivariate analysis adjusted for age, sex, current smoking status, alcohol consumption and lipid medication

* p value was statistically significant.

i. Data are medians. The data in parentheses are the first and third quartiles.

higher total skeletal muscle CSA (p = 0.041) than MHO individuals in the model that adjusted for all variables. However, no significant difference existed in total skeletal muscle SMI, SMD, NAM index, LAM index and IMAT index in all adjusted model (Model 3).

### Correlation between muscle imaging parameters and number of metabolic syndrome components

Correlations between muscle imaging parameters and number of metabolic syndrome components are presented in Table 4. In women, total skeletal muscle CSA (r = 0.269, p = 0.004) and SMI (r = 0.301, p = 0.001) (i.e., muscle quantity) were positively associated with number of metabolic syndrome components. In men and women, no association was found between SMD (i.e., muscle quality) and number of metabolic syndrome components. No significant correlation existed between intramuscular compositional indices (e.g., NAM index, LAM index, and IMAT index) and number of metabolic syndrome components in men and women

## Discussion

The purpose of our study was to determine the differences in the quantity and quality of skeletal muscle between MHO and MUO individuals using CT to demonstrate the existence of

**Table 4. Correlations between muscle imaging parameters and the number of metabolic syndrome components in MHO and MUO participants.**

| Variable | Correlation with number of metabolic syndrome components | | | | | |
|---|---|---|---|---|---|---|
| | Total (n = 172) | | Men (n = 53) | | Women (n = 119) | |
| | r | p value | r | p value | r | p value |
| CSA ($cm^2$) | 0.174 | 0.023* | -0.096 | 0.493 | 0.269 | 0.004* |
| SMI [$cm^2/(m^2)$] | 0.185 | 0.016* | -0.197 | 0.157 | 0.301 | 0.001* |
| SMD (HU) | 0.053 | 0.493 | 0.110 | 0.434 | 0.025 | 0.786 |
| NAM index [$cm^2/(m^2)$] | 0.097 | 0.210 | -0.116 | 0.408 | 0.142 | 0.127 |
| LAM index [$cm^2/(m^2)$] | 0.084 | 0.276 | -0.130 | 0.352 | 0.150 | 0.108 |
| IMAT index [$cm^2/(m^2)$] | -0.054 | 0.489 | -0.210 | 0.132 | -0.008 | 0.935 |

Note. CSA, cross-sectional area; SMI, skeletal muscle index; SMD, skeletal muscle density; HU, Hounsfield unit; NAM, normal attenuation muscle; LAM, low-attenuation muscle; IMAT, intermuscular adipose tissue.

* p value was statistically significant.

specific subtypes among people with morbid obesity such as MHO and MUO. In addition, we investigated whether the intramuscular composition, classified as NAM, LAM, and IMAT indices, were significantly different between MHO and MUO individuals. Our findings suggested that total skeletal muscle SMI (i.e., muscle quantity), SMD (i.e., muscle quality), and intramuscular compositional indices such as NAM index, LAM index, and IMAT index did not differ between MHO and MUO individuals after adjustment for age, sex, current smoking status, alcohol consumption and lipid medication. This finding suggested that ectopic fat role is limited in people with morbid obesity with metabolic abnormalities, at least from the skeletal muscular point of view.

When metabolic syndrome develops, ectopic fat infiltration increases. Pieńkowska et al. [32] showed that, during metabolic syndrome development, the most rapid fat infiltration site is the muscle. This finding indicates that muscle could be rapidly affected by metabolic risk. However, in the present study, skeletal muscle fat infiltration in obesity without metabolic risk did not differ from obesity with metabolic risk, after adjusting for age, sex and other covariate factors. Taken together, metabolically healthy obesity could be a transient phenotype leading to metabolically unhealthy obesity, which results in an increased risk of cardiovascular disease and type 2 diabetes mellitus [5, 33, 34]. In contrast to our study, Stefan et al. [15] reported that muscle fat infiltration was lower in MHO individuals than in MUO individuals on proton (hydrogen 1 [$^1$H]) magnetic resonance (MR) spectroscopy. However, they showed that fat deposition in skeletal muscle was significantly different between MHO and MUO individuals only for the tibialis anterior muscle, and that muscle accounts for a small portion of the whole-body muscle mass. Moreover, $^1$H MR spectroscopy is usually conducted using a single-voxel technique, and only one lesion (of approximately 1 cm$^3$) can be examined, thus limiting the evaluation of an entire muscle. We analyzed all skeletal muscles (i.e., paraspinal, psoas, quadratus lumborum, transversus abdominis, rectus abdominis and internal and external obliques) at the L3 level. Kim et al. [26] demonstrated that unlike our studies, the quality and quantity of skeletal muscle were significantly different between MHO and MUO participants. Despite of large number of study participants, there is a big difference from our cohort. Previous study included people who visited the health screening center for regular medical checkups, and the average BMI of MHO and MUO participants was 26.3 and 27.3, respectively. This population could not represent the participant with obesity. In our study, we targeted people with morbid obesity, and the average BMI of MHO and MUO participants was approximately 40.

Skeletal muscle is the major source of glucose disposal, and unlike muscle mass increase that comes through exercise, increased skeletal muscle mass in obesity is associated with decreased insulin sensitivity [35]. Decreased insulin sensitivity and chronic elevation of insulin levels have been found to directly influence skeletal muscle mass by stimulating contractile protein synthesis in animal studies [36, 37]. Other studies have also demonstrated that high levels of insulin can enhance muscle hypertrophy and that increased muscle mass as a result of obesity is not associated with muscle strength [38]. These studies' findings support our results, which showed that SMI (i.e., muscle quantity) was associated with increased metabolic risk, and SMD (i.e., muscle quality) was not associated with increased metabolic risk.

Skeletal muscle fibers can be broadly categorized into two types: 'slow-twitch' (type 1) and 'fast-twitch' (type 2) [39]. Fast-twitch fibers are further divided into three main subtypes: types 2A, 2X, and 2B. Type 1 and 2A fibers primarily utilize oxidative metabolism, while type 2X and 2B fibers mainly depend on glycolytic metabolism [39]. We conducted an analysis on predominantly postural muscles, which contain a higher proportion of type 1 fibers [40]. Talbot et al. found that both obese individuals and individuals with type 2 diabetes mellitus exhibited a decreased percentage of type 1 muscle fibers, and this percentage of type 1 fibers was found to be correlated with insulin sensitivity [39]. Because of this, we feel our study reflects insulin

sensitivity in skeletal muscle better than studies that evaluated skeletal muscle quantity and quality using CT images of the limbs, and this is a strength of our study.

We also evaluated intramuscular compositional indices such as the LAM index, NAM index, and IMAT index in MHO and MUO individuals. LAM is associated with lipid-rich skeletal muscle, in which lipid stores are contained between and within the muscle [41]. Tanaka et al. [42] demonstrated that the LAM index or SMD was associated with a high incidence of diabetes mellitus, whereas the NAM index or SMI, was not. In our study, the LAM index did not significantly differ between MHO and MUO individuals. These results suggest that metabolically healthy obesity is a transient phenotype during the progression of obesity complications. In addition, our study included people with morbid obesity who required bariatric surgery. The overwhelming amount of fat in our group could have ameliorated the difference of skeletal ectopic fat, according to metabolic phenotype.

Compared to MUO individuals, MHO individuals are characterized by preserved insulin sensitivity and specific body fat distributions such as more subcutaneous fat depots, lower visceral fat and less ectopic fat accumulation in the liver and skeletal muscle, less macrophage infiltration and inflammation in adipose tissue, and higher levels of adiponectin, the most abundant protein secreted by adipose tissue [4, 14, 43–45]. However, several meta-analyses of prospective cohort studies have demonstrated that most MHO individuals have a significantly increased risk of cardiovascular disease and type 2 diabetes mellitus compared to healthy normal weight individuals [5, 33, 34]. Controversy remains regarding the existence of a distinctive subtype of obesity; however, no quantitative analysis has proven this. To the best of our knowledge, this report is the first to assess ectopic fat accumulation in skeletal muscle in MHO and MUO individuals using CT-based quantitative analysis.

Our study has several limitations. First, this study was a retrospective study conducted at a single institution and was confined to Korean men and women. These factors have the potential of causing selection bias and limit the ability to generalize the findings to other ethnic groups. Second, we analyzed skeletal muscles using a single slice at the L3 level and the results may not reflect whole-body skeletal muscle. Furthermore, IMAT is known to be unevenly distributed [46]. Further prospective studies of skeletal muscle throughout the whole body are needed. Third, this study included people with morbid obesity who needed bariatric surgery. Therefore, a selection bias may exit. Fourth, in terms of baseline characteristics, MUO individuals were older and heavier than MHO individuals, limiting our ability to conduct proper comparisons between the MUO and MHO individuals. This is because age and weight are major factors affecting muscle [47–49]. Fifth, we did not evaluate muscle function (i.e. strength, muscle endurance) and only assessed muscle quality through structural changes. Further research is needed on muscle quality which takes into account the function and structure of muscles.

## Conclusion

In conclusion, the quality, quantity, and intramuscular composition of the total skeletal muscles at the L3 level did not differ between MHO and MUO individuals. This finding suggests that the contribution of skeletal muscle to metabolic healthy state may be limited in the people with morbid obesity. In addition, MHO individuals who are considered "healthy" should be carefully monitored and can have a similar risk of metabolic complications as MUO individuals, at least based on an assessment of skeletal muscle.

## Supporting information

**S1 Table. STOBE statement checklist.**
(DOCX)

**S1 Data. Reliability of muscle imaging parameter measurement.**
(ZIP)

## Author Contributions

**Conceptualization:** Eunsun Oh, Sang Hyun Kim, Hyeong Kyu Park, Soon Hyo Kwon.

**Data curation:** Sang Hyun Kim, Soon Hyo Kwon.

**Formal analysis:** Nam-Jun Cho, Heemin Kang.

**Funding acquisition:** Eunsun Oh, Soon Hyo Kwon.

**Investigation:** Heemin Kang, Soon Hyo Kwon.

**Methodology:** Nam-Jun Cho, Heemin Kang.

**Project administration:** Sang Hyun Kim.

**Resources:** Sang Hyun Kim, Hyeong Kyu Park.

**Software:** Eunsun Oh, Heemin Kang.

**Supervision:** Hyeong Kyu Park, Soon Hyo Kwon.

**Visualization:** Eunsun Oh, Nam-Jun Cho.

**Writing – original draft:** Eunsun Oh, Nam-Jun Cho.

**Writing – review & editing:** Eunsun Oh, Nam-Jun Cho, Sang Hyun Kim, Hyeong Kyu Park, Soon Hyo Kwon.

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
