## [Decision Letter · Decision Letter 0]

29 Sep 2023

PONE-D-23-20750Computed tomography evaluation of skeletal muscle quality and quantity in morbidly obese individuals with and without metabolic abnormalityPLOS ONE

Dear Dr. Soon Hyo Kwon,

Thank you for submitting your manuscript to PLOS ONE. After careful consideration, we feel that it has merit but does not fully meet PLOS ONE’s publication criteria as it currently stands. Therefore, we invite you to submit a revised version of the manuscript that addresses the points raised during the review process.

We look forward to receiving your revised manuscript.

Kind regards,

Valeria Guglielmi

Academic Editor

PLOS ONE

Journal Requirements:

"S. H. K. received the funding from Soonchunhyang University Research Fund and the Korean government (MSIT) (NRF-2019M3E5D3073102).

E. O. received the National Research Foundation of Korea (NRF) grant funded by the Ministry of Education, Science and Technology (NRF-2019R1G1A1002919)."

Reviewers' comments:

Reviewer's Responses to Questions

**Comments to the Author**

1. Is the manuscript technically sound, and do the data support the conclusions?

Reviewer #1: Partly

Reviewer #2: Yes

Reviewer #3: Partly

2. Has the statistical analysis been performed appropriately and rigorously? 

Reviewer #1: No

Reviewer #2: Yes

Reviewer #3: Yes

3. Have the authors made all data underlying the findings in their manuscript fully available?

Reviewer #1: No

Reviewer #2: Yes

Reviewer #3: Yes

4. Is the manuscript presented in an intelligible fashion and written in standard English?

Reviewer #1: Yes

Reviewer #2: Yes

Reviewer #3: Yes

5. Review Comments to the Author

Reviewer #1: These authors utilise data from 236 individuals to ascertain whether there are differences in the quantity (anatomical CSA) and inferred quality (adiposity/fatty infiltration) of skeletal muscle between metabolically health vs metabolically unhealthy obese adults. They also aimed to clarify whether intramuscular composition (grouped in 3 classes) would be significantly different between these individuals.

The article is written without any major prose issues, and the author used a technique (i.e. CT) previously used by others to infer similar conclusions on muscle content and quality. Nonetheless there are a few points that need addressing.

Abstract

1- Line 7- please specify if this is anatomical CSA or physiological CSA

2- Th abstract requires a statement summarising the relevance of this work

INTRODUCTION

1- Given this paper in on skeletal muscle function and characteristics, it is important that a more complete synthesis of the literature that precedes this work is carried out. I would recommend the following work on the impact of obesity on skeletal muscle quantity and quality

a. Erskine RM, Tomlinson DJ, Morse CI, Winwood K, Hampson P, Lord JM, and Onambele GL. The individual and combined effects of obesity- and ageing-induced systemic inflammation on human skeletal muscle properties. Int J Obes (Lond) 2017;41:102-11

b. Tomlinson DJ, Erskine RM, Morse CI, Winwood K, and Onambélé GL. Combined effects of body composition and ageing on joint torque, muscle activation and co-contraction in sedentary women. Age (Dordr) 2014;36:9652 Erratum in 2014;36:9662

c. Tomlinson DJ, Erskine RM, Morse CI, Winwood K, and Onambélé GL. Obesity decreases both whole muscle and fascicle strength in young females but only exacerbates the ageing-related whole muscle level asthenia. Physiol Rep 2014;2:e12030.

2- Please state a hypothesis at the end of the introduction

METHODS

1- Please provide the reliability of all measures as either coefficients of variation, or Typical error and systematic error (ideally all 3). In addition, can you also provide the compatibility intervals for the provided ICCs values, as well as the coefficient of variance or standard error or systematic error of all measurements especially muscle quantity and quality.

2- What is unclear, is whether the method of using CT scans has been currently or previously externally validated against a gold standard and what the results of such a validation are. These need to be stated here. Indeed : it is hard to interpret how ICC translates to actual measurement error.

3- Statistical analysis:

a. please specify the approach to determine normal distribution

b. it is unclear whether equal variance was also determined prior to running ANOVAs. If so, this must be stated here and what corrections were applied for non-equal distributions

c. For the multiple linear regressions, (1) what model did you use and why (e.g. forced entry, forward, backward etc?), (2) did you check for collinearity, (3) make a comment regrading whether you considered checking the normal distribution of the residuals.

RESULTS

1- The fact that MUO individuals were heavier and older is a very significant factor to consider in the between group comparisons (see ref). i.e. there is an argument here for the necessity for ANCOVAs to be run, not just ANOVAs, and partial correlations (not simple correlations)

Ref: Tomlinson DJ, Erskine RM, Morse CI, Winwood K, and Onambélé GL. The impact of obesity on skeletal muscle architecture in untrained young vs. old women. J Anat 2014;225:675-84

2- Given the fact that body composition in men and women would not necessarily have the same muscle impact, this reviewer would argue that models and comparisons ought to be segregated by sex not in a pooled sample (please see references suggested in other comments)

DISCUSSION

This reviewer did not delved too deeply in the discussion in view of some of the issues that they felt first need to be addressed in the statistical approaches and hence results.

Reviewer #2: The authors compard skeletal muscle quality and quantity in morbidly obese invidiuals with and without abnormality. They did the measurements via. CT scan at L3 level.

Short title should be shortened.

The authors made the measurements at only L3 level, however it has previously been demonstrated that antropometric values correlate mostly with L1-2 and L2-3 level measurements. Even BMI has been shown to be an unreliable antropometric parameter. Besides, it could not depict any difference between women and men, as we all know, the two gender have different antropometric properties. Instead, a very known parameter has been transformed to depict any association between obesity and spine degeneration: SFI (Berikol G, Ekşi MŞ, Aydın L, Börekci A, Özcan-Ekşi EE: Subcutaneous fat index: a reliable tool for lumbar spine studies. Eur Radiol 2022, 32(9): 6504-6513.)

In the conclusion section, there is a sentence like this: ‘We propose that MHO

individuals who are considered “healthy” cannot truly be healthy and can have a similar risk

of metabolic complications as MHO individuals, at least based on an assessment of skeletal

muscle.’ Could it be ‘…. complications as MUO individuals…’?

Reviewer #3: Oh et al, in their study aimed to compare the muscle quantity and quality between metabolically healthy and unhealthy people with obesity through the usage of CT imaging in the abdominal area.

Even when the study is well-planned and executed, and the results are quite interesting and should be published, there are some sections that need review and analysis from the authors before do so.

Abstract:

It is not clear here why a healthy control group was incorporated into the study. Please add some information on this regard.

Introduction

Is there a clear definition of what a metabolically healthy person is? Maybe regarding his/her insulin sensitivity?

Methods

Please change the term morbidly obese patients for people with morbid obesity.

The fact that measurements were conducted using three different scanners, could derived in possible variability inter-instruments?

Why the authors did not used the clinical data regarding insulin sensitivity (such as fasting glucose, fasting insulin, HOMA-IR, HbA1c, OGTT) to classify the MUO and MHO groups?

Why the correlations were performed between the imaging outcomes and the number of MS components, and not versus the glucose metabolism parameters, which have been described as closely related with muscle quality? From my perspective, there an opportunity lost there.

Discussion

The fact that the group MUO was heavier and older should be considered a limitation for proper comparisons between groups, as age and weight are major factors regarding muscle structure and function, even when on the multivariate analysis, the models corrected by these factors.

Muscle quality not only depends on architectural changes, but also is associated with muscle function (i.e. strength, muscle endurance). Therefore, the authors should be careful when discussing the absence of differences between groups in terms of muscle quality. From the results from this study, it is clear that there are no differences in intramuscular fat infiltration between MHO and MUO, however, with the absence of muscle function data, statements about muscle quality are problematic.

Please discuss regarding the fact that mainly postural muscles were analyzed (which have a higher level of oxidative fibers), whereas oxidative-glycolytic muscle such as quadriceps could have a different pattern regarding the outcomes of this study.

Conclusion

Careful with statements like this: “We propose that MHO individuals who are considered “healthy” cannot truly be healthy” because there are several research groups that have stated that MHO persons have in fact a metabolic function similar/equal to people with normal weight and metabolic function. A different thing is the increased risk of moving from a MHO to a MUO phenotype.

6. PLOS authors have the option to publish the peer review history of their article (what does this mean?). If published, this will include your full peer review and any attached files.

Reviewer #1: No

Reviewer #2: No

Reviewer #3: **Yes: **Sergio Martinez-Huenchullan

---

## [Author Response · Author response to Decision Letter 0]

15 Nov 2023

Response to Reviewers

Reviewer #1: 

These authors utilise data from 236 individuals to ascertain whether there are differences in the quantity (anatomical CSA) and inferred quality (adiposity/fatty infiltration) of skeletal muscle between metabolically health vs metabolically unhealthy obese adults. They also aimed to clarify whether intramuscular composition (grouped in 3 classes) would be significantly different between these individuals.

The article is written without any major prose issues, and the author used a technique (i.e. CT) previously used by others to infer similar conclusions on muscle content and quality. Nonetheless there are a few points that need addressing.

Abstract

1- Line 7- please specify if this is anatomical CSA or physiological CSA

; Thank you for your comment. We agree with your opinion. We have revised the manuscript to specify that it is anatomical CSA.

2- The abstract requires a statement summarising the relevance of this work

; Thank you for your comment. As you suggested, we have revised the abstract.

INTRODUCTION

1- Given this paper in on skeletal muscle function and characteristics, it is important that a more complete synthesis of the literature that precedes this work is carried out. I would recommend the following work on the impact of obesity on skeletal muscle quantity and quality

a. Erskine RM, Tomlinson DJ, Morse CI, Winwood K, Hampson P, Lord JM, and Onambele GL. The individual and combined effects of obesity- and ageing-induced systemic inflammation on human skeletal muscle properties. Int J Obes (Lond) 2017;41:102-11

b. Tomlinson DJ, Erskine RM, Morse CI, Winwood K, and Onambélé GL. Combined effects of body composition and ageing on joint torque, muscle activation and co-contraction in sedentary women. Age (Dordr) 2014;36:9652 Erratum in 2014;36:9662

c. Tomlinson DJ, Erskine RM, Morse CI, Winwood K, and Onambélé GL. Obesity decreases both whole muscle and fascicle strength in young females but only exacerbates the ageing-related whole muscle level asthenia. Physiol Rep 2014;2:e12030.

; Thank you for your comment. We have reviewed the papers you mentioned and added them to the list of references.

2- Please state a hypothesis at the end of the introduction

; Thank you for your comment. As you suggested, we have presented a hypothesis at the end of the introduction.

METHODS

1- Please provide the reliability of all measures as either coefficients of variation, or Typical error and systematic error (ideally all 3). In addition, can you also provide the compatibility intervals for the provided ICCs values, as well as the coefficient of variance or standard error or systematic error of all measurements especially muscle quantity and quality.

; Thank you for your comment. As you suggested, we have presented the coefficient of variation, typical error and systematic error for all measure on the supplementary Table 2. Typical error = sqrt(mean((x-y)^2)/2), which we defined as averaging the squares of the two people's differences, dividing by 2, and taking the root. We defined systematic error = |mean(x-y)|, which is the absolute value of the mean of the two people's differences.

2- What is unclear, is whether the method of using CT scans has been currently or previously externally validated against a gold standard and what the results of such a validation are. These need to be stated here. Indeed : it is hard to interpret how ICC translates to actual measurement error.

; Thank you for your comment. There are many studies on the assessment of muscle quality by CT or magnetic resonance imaging and we have referenced the following studies:

1. Hamrick MW, McGee-Lawrence ME, Frechette DM. Fatty infiltration of skeletal muscle: mechanisms and comparisons with bone marrow adiposity. Front Endocrinol (Lausanne). 2016;7:69.

2. Aubrey J, Esfandiari N, Baracos VE, et al. Measurement of skeletal muscle radiation attenuation and basis of its biological variation. Acta Physiol (Oxf). 2014;210(3):489-497.

3. Goodpaster BH, Thaete FL, Kelley DE. Composition of skeletal muscle evaluated with computed tomography. Ann N Y Acad Sci. 2000;904:18-24.

4. Addison O, Marcus RL, Lastayo PC, Ryan AS. Intermuscular fat: a review of the consequences and causes. Int J Endocrinol. 2014;2014:309570. doi:10.1155/2014/309570

5. Amini B, Boyle SP, Boutin RD, Lenchik L. Approaches to assessment of muscle mass and myosteatosis on computed tomography: a systematic review. J Gerontol A Biol Sci Med Sci. 2019;74(10):1671-1678.

3- Statistical analysis:

a. please specify the approach to determine normal distribution

; Thank you for your comment. The distribution of the variables was visually assessed through a histogram and quantile-quantile plot. We have added this information to the revised manuscript.

b. it is unclear whether equal variance was also determined prior to running ANOVAs. If so, this must be stated here and what corrections were applied for non-equal distributions

; Thank you for your comment. We have made some revisions to the Methods section. Initially, we used ANOVA or the Kruskal–Wallis test to compare the three groups simultaneously. However, in the latest revision, we compared the means of only two groups (the MHO and MUO groups). For this purpose, t-tests or Mann-Whitney U tests were employed depending on the circumstances. We have checked the assumption of equal variances between two groups by Levene’s test. If this assumption was met, we utilized Student's t-test; otherwise, we applied Welch's t-test. We have modified the content as follows. 

“The comparisons between MHO group and MUO group were analyzed using Student’s t-test, Welch's t-test or Mann-Whitney U test, as appropriate. We used Levene's test to assess the equality of variances between the two groups.”

c. For the multiple linear regressions, (1) what model did you use and why (e.g. forced entry, forward, backward etc?), 

; Thank you for your comment. Regression modeling is performed for various reasons. It may aim to create a model that predicts a specific outcome or to investigate causality between particular variables and the outcome. Variable selection by statistical methods such as backward elimination, forward selection, best subset selection, or LASSO may be more suitable for prediction modeling. However, we applied multiple linear regression to determine the independent effect of metabolic syndrome on muscle quality and quantity in obese patients. In such situations, it is more appropriate to select variables that may influence the outcome based on existing knowledge. So, we selected age, sex, current smoking status, alcohol consumption, and lipid medication as the variables for adjustment.

(2) did you check for collinearity, 

; There were no high correlations among the explanatory variables that would raise concerns about multicollinearity.

(3) make a comment regrading whether you considered checking the normal distribution of the residuals.

; We have checked the residuals by Residuals versus fitted plot and Quantile-quantile plot. The residuals derived from linear regression models were randomly scattered and normally distributed, excluding the IMAT index. After log transformation of the IMAT index, the residuals of the regression model of the IMAT index were also normally distributed.

RESULTS

1- The fact that MUO individuals were heavier and older is a very significant factor to consider in the between group comparisons (see ref). i.e. there is an argument here for the necessity for ANCOVAs to be run, not just ANOVAs, and partial correlations (not simple correlations)

Ref: Tomlinson DJ, Erskine RM, Morse CI, Winwood K, and Onambélé GL. The impact of obesity on skeletal muscle architecture in untrained young vs. old women. J Anat 2014;225:675-84

; Thank you for your comment. In the baseline characteristics table (Table 1), it was more appropriate to present only the simple differences between the two groups. So, we used the t-test or Mann-Whitney U test to compare some parameters between the two groups rather than covariate adjustment approaches. In Table 3, we used multiple linear regression to assess the differences in CT muscle parameters between the two groups while adjusting confounding variables. Multiple linear regression can be a suitable alternative to ANCOVA when adjusting for multiple variables.

2- Given the fact that body composition in men and women would not necessarily have the same muscle impact, this reviewer would argue that models and comparisons ought to be segregated by sex not in a pooled sample (please see references suggested in other comments)

; Thank you for your comment. We agree with your opinion. In Tables 2 and 4, we have already conducted separate analyses for males and females, and in Table 3, we have applied gender as an adjustment variable. In our multiple linear regression models, gender was significantly associated with CT muscle parameters, as expected.

DISCUSSION

This reviewer did not delved too deeply in the discussion in view of some of the issues that they felt first need to be addressed in the statistical approaches and hence results.

; We have added to our Discussion section. Your comment has improved our manuscript. Thank you.

Reviewer #2:

 The authors compard skeletal muscle quality and quantity in morbidly obese invidiuals with and without abnormality. They did the measurements via. CT scan at L3 level.

# Short title should be shortened.

; Thank you for your comment. We agree with your opinion.

# The authors made the measurements at only L3 level, however it has previously been demonstrated that antropometric values correlate mostly with L1-2 and L2-3 level measurements. Even BMI has been shown to be an unreliable antropometric parameter. Besides, it could not depict any difference between women and men, as we all know, the two gender have different antropometric properties. Instead, a very known parameter has been transformed to depict any association between obesity and spine degeneration: SFI (Berikol G, Ekşi MŞ, Aydın L, Börekci A, Özcan-Ekşi EE: Subcutaneous fat index: a reliable tool for lumbar spine studies. Eur Radiol 2022, 32(9): 6504-6513.)

; In the research setting, a single cross-sectional abdominal image from a diagnostic CT has been used to assess abdominal circumference, abdominal adipose tissue and skeletal muscle areas. A single cross-sectional CT image of the L3 region can accurately estimate body composition, including regional abdominal adipose tissue, skeletal muscle and waist circumference. We have added these sentences to the Methods section. We referenced the following paper: 

Shen W, Punyanitya M, Wang Z, et al. Total body skeletal muscle and adipose tissue volumes: estimation from a single abdominal cross-sectional image. Journal of applied physiology (Bethesda, Md: 1985) 2004;97(6):2333–2338.

# In the conclusion section, there is a sentence like this: ‘We propose that MHO individuals who are considered “healthy” cannot truly be healthy and can have a similar risk of metabolic complications as MHO individuals, at least based on an assessment of skeletal muscle.’ Could it be ‘…. complications as MUO individuals…’?

; Thank you for your comment. As you and reviewer #3 have suggested, we have revised the Conclusion section.

Reviewer #3: 

Oh et al, in their study aimed to compare the muscle quantity and quality between metabolically healthy and unhealthy people with obesity through the usage of CT imaging in the abdominal area.

Even when the study is well-planned and executed, and the results are quite interesting and should be published, there are some sections that need review and analysis from the authors before do so.

Abstract:

# It is not clear here why a healthy control group was incorporated into the study. Please add some information on this regard.

; Thank you for your comment. This has improved our manuscript. We thought that a healthy control group could provide a reference level for skeletal muscle quality. Participants with obesity where assigned a metabolic phenotype of healthy when their skeletal muscle quality of did not differ from that of the healthy control group. We have added sentences to this effect to the Abstract. 

Introduction

# Is there a clear definition of what a metabolically healthy person is? Maybe regarding his/her insulin sensitivity?

; Thank you for your comment. Our study is a retrospective analysis, and we did not have data on insulin sensitivity. However, many studies define metabolic health based on absence of the metabolic syndrome. Previously, we have reported on perivascular fat differences based on the same definition.

1. Tsatsoulis A, Paschou SA. Metabolically Healthy Obesity: Criteria, Epidemiology, Controversies, and Consequences. Curr Obes Rep. 2020 Jun;9(2):109-120. doi: 10.1007/s13679-020-00375-0. PMID: 32301039.

2. Lee EJ, Cho NJ, Kim H, Nam B, Jeon JS, Noh H, Han DC, Kim SH, Kwon SH. Abdominal periaortic and renal sinus fat attenuation indices measured on computed tomography are associated with metabolic syndrome. Eur Radiol. 2022 Jan;32(1):395-404. doi: 10.1007/s00330-021-08090-7. Epub 2021 Jun 22. PMID: 34156551. 

Methods

# Please change the term morbidly obese patients for people with morbid obesity.

; Thank you for your comment. As you suggested, we have revised the manuscript.

# The fact that measurements were conducted using three different scanners, could derived in possible variability inter-instruments?

; Thank you for your comment. We thought the likelihood of variability in images obtained using different CT machines was likely to be very low. In the clinical setting, we also used different CT machines on the same patient at follow-up. Although the images were obtained from different CT machines, we analyzed them with the same software program. 

# Why the authors did not used the clinical data regarding insulin sensitivity (such as fasting glucose, fasting insulin, HOMA-IR, HbA1c, OGTT) to classify the MUO and MHO groups?

; Thank you for your comment. Our study is a retrospective analysis, and we did not have data on insulin sensitivity. 

# Why the correlations were performed between the imaging outcomes and the number of MS components, and not versus the glucose metabolism parameters, which have been described as closely related with muscle quality? From my perspective, there an opportunity lost there.

; Our study aims to evaluate whether skeletal muscle quality measured by CT could be differentiated according to metabolic health status. MHO has been considered to have better health outcomes compare to MUO. Many studies commonly defined metabolic health based on absence of the metabolic syndrome. Previous studies showed that muscle quality measured by CT provided prognostic value in various cohorts. Therefore, we hypothesized that the quality and quantity of skeletal muscle in MHO would be different from that in MUO. 

Discussion

# The fact that the group MUO was heavier and older should be considered a limitation for proper comparisons between groups, as age and weight are major factors regarding muscle structure and function, even when on the multivariate analysis, the models corrected by these factors.

; Thank you for your comment. We agree with your opinion. We have added that information to the discussion of limitations of the study, as you suggested.

# Muscle quality not only depends on architectural changes, but also is associated with muscle function (i.e. strength, muscle endurance). Therefore, the authors should be careful when discussing the absence of differences between groups in terms of muscle quality. From the results from this study, it is clear that there are no differences in intramuscular fat infiltration between MHO and MUO, however, with the absence of muscle function data, statements about muscle quality are problematic.

; Thank you for your comment. We agree with your opinion. We have added that information to the discussion of the limitations of the study, as you suggested.

# Please discuss regarding the fact that mainly postural muscles were analyzed (which have a higher level of oxidative fibers), whereas oxidative-glycolytic muscle such as quadriceps could have a different pattern regarding the outcomes of this study.

; Thank you for your comment. We agree with your opinion. We added that information to the Discussion section.

Conclusion

# Careful with statements like this: “We propose that MHO individuals who are considered “healthy” cannot truly be healthy” because there are several research groups that have stated that MHO persons have in fact a metabolic function similar/equal to people with normal weight and metabolic function. A different thing is the increased risk of moving from a MHO to a MUO phenotype.

; Thank you for your comment. We agree with your opinion. We have tone downed our statement and we have revised the Conclusion section.

---

## [Decision Letter · Decision Letter 1]

6 Dec 2023

Computed tomography evaluation of skeletal muscle quality and quantity in people with morbid obesity with and without metabolic abnormality

PONE-D-23-20750R1

Dear Dr. Kwon,

We’re pleased to inform you that your manuscript has been judged scientifically suitable for publication and will be formally accepted for publication once it meets all outstanding technical requirements.

Kind regards,

Valeria Guglielmi

Academic Editor

PLOS ONE

Additional Editor Comments (optional):

Reviewers' comments:

Reviewer's Responses to Questions

**Comments to the Author**

1. If the authors have adequately addressed your comments raised in a previous round of review and you feel that this manuscript is now acceptable for publication, you may indicate that here to bypass the “Comments to the Author” section, enter your conflict of interest statement in the “Confidential to Editor” section, and submit your "Accept" recommendation.

Reviewer #1: All comments have been addressed

Reviewer #2: (No Response)

Reviewer #3: All comments have been addressed

2. Is the manuscript technically sound, and do the data support the conclusions?

Reviewer #1: Yes

Reviewer #2: Partly

Reviewer #3: Yes

3. Has the statistical analysis been performed appropriately and rigorously? 

Reviewer #1: Yes

Reviewer #2: Yes

Reviewer #3: Yes

4. Have the authors made all data underlying the findings in their manuscript fully available?

Reviewer #1: Yes

Reviewer #2: Yes

Reviewer #3: Yes

5. Is the manuscript presented in an intelligible fashion and written in standard English?

Reviewer #1: Yes

Reviewer #2: Yes

Reviewer #3: Yes

6. Review Comments to the Author

Reviewer #1: The authors have systematically addressed all comments/recommendations i had in my previous review. This reviewer has no further comment.

Reviewer #2: the authors answered most of the queries. however, there is a fact about inappropriateness of selecting of only one level for evaluation of whole lumbar spine. In the paper by Berikol et al. the authors have already depicted that all levels are valuable indicators for anthropometric represantation of the subjects, yet L1-l2 is the most suitable one. so the readers should be warned about this proven fact by the authors in their limitations section.

Reviewer #3: Dear authors,

thank you for considering my comments and suggestions.

I don't have further comments.

Kind regards.

7. PLOS authors have the option to publish the peer review history of their article (what does this mean?). If published, this will include your full peer review and any attached files.

Reviewer #1: No

Reviewer #2: No

Reviewer #3: **Yes: **Sergio Martinez-Huenchullan

---

## [Editor Report · Acceptance letter]

13 Dec 2023

PONE-D-23-20750R1 

PLOS ONE

Dear Dr. Kwon, 

I'm pleased to inform you that your manuscript has been deemed suitable for publication in PLOS ONE. Congratulations! Your manuscript is now being handed over to our production team.

Kind regards, 

on behalf of

Prof. Valeria Guglielmi 

Academic Editor

PLOS ONE